# A Sex- and Gender-Based Approach to Chronic Conditions in Central Catalonia (Spain): A Descriptive Cross-Sectional Study

**DOI:** 10.3390/ijerph21020152

**Published:** 2024-01-29

**Authors:** Georgina Pujolar-Díaz, Queralt Miró Catalina, Aïna Fuster-Casanovas, Laia Sola Reguant, Josep Vidal-Alaball

**Affiliations:** 1Unitat de Suport a la Recerca de la Catalunya Central, Fundació Institut Universitari per a la Recerca a l’Atenció Primària de Salut Jordi Gol i Gurina, 08272 Sant Fruitos de Bages, Spain; qmiro.cc.ics@gencat.cat (Q.M.C.); afuster.cc.ics@gencat.cat (A.F.-C.); lsola.cc.ics_ext@gencat.cat (L.S.R.); jvidal.cc.ics@gencat.cat (J.V.-A.); 2Health Promotion in Rural Areas Research Group, Gerència d’Atenció Primària i a la Comunitat de la Catalunya Central, Institut Català de la Salut, 08272 Sant Fruitos de Bages, Spain; 3Faculty of Medicine, Universitat de Vic—Universitat de Catalunya Central, 08500 Vic, Spain

**Keywords:** chronic conditions, gender inequality, gendered approach, primary healthcare, multimorbidity

## Abstract

The growth of chronic conditions worldwide poses a challenge for both health systems and the quality of life of people with these conditions. However, sex- and gender-based approaches are scarce in this field. Adopting this perspective, this study aims to describe the prevalence of chronic conditions in the Bages–Moianès region (Catalonia, Spain), and analyse the associations of chronic conditions with sex and age. This cross-sectional study used data from the population assigned to the Catalan Health Institute primary care settings in this area between 2018 and 2021 (*n* = 163,024). A total of 26 chronic conditions (grouped into 7 typologies), sex and age were the analysis variables. A total of 75,936 individuals presented at least one chronic condition, representing 46.6% of the analysed population. The prevalence was higher among women and older individuals. Being male was associated with a greater probability of presenting cardiovascular diseases, neurodevelopmental disorders and metabolic diseases and a lower probability of presenting neurodegenerative diseases, chronic pain and mental health disorders. Adjusting by sex, a positive age gradient was observed in most groups, except for respiratory diseases and mental health disorders. Chronic conditions have a high prevalence in the Bages–Moianès region, showing differences in typology, sex and age. Adopting gender perspectives (both in health systems and future research) is crucial when dealing with chronic conditions in order to take into account their differential impact.

## 1. Introduction

Social and health progress has allowed the life expectancy to increase over the past few decades worldwide [1,2]. However, a longer life has also meant an increased risk of morbidity, in addition to making the prevalence of chronic conditions increasingly high [3]. At the same time, chronic multimorbidity, which is defined as the presence of more than one chronic condition [4,5], is especially common among the older population [2,3,4,5,6,7]. This increase in the burden of chronic conditions [3] poses a challenge not only in the approach that health systems must face in terms of care and costs [2,7,8] but also in relation to the loss of quality of life of people with these conditions and the needs they develop in this regard [2,6,7].

Despite this increasing pattern of chronic conditions, there are differences in how these diseases are distributed among men and women, besides the diversity of their symptomatology and outcomes. However, the existence of studies on this differential impact is still limited [9,10,11,12]. Health systems respond to traditionally patriarchal and androcentric knowledge, based on men’s bodies and needs, which has allowed centuries of inattention to and ignorance of women’s characteristics and illnesses [13,14,15]. From this perspective, medicine has historically disregarded women’s health needs, often determining them as atypical or characterising them as psychiatric, which has contributed to marginalising women [15,16,17]. Although women are now included in clinical research, their systematic exclusion underlies a lack of information and data on differences in symptom presentation, diagnoses and response to treatments [10,15,18,19,20].

This lack of knowledge has been replicated in the approach to health and disease, thus influencing the care received and the quality of life derived from this [10,15,21]. In this regard, it can make it difficult to address women’s health problems accurately and in a timely manner, as well as causing delays in obtaining proper diagnosis and treatment [15]. However, there is a lack of funding to develop studies in these fields, which perpetuates gender bias as a barrier to progress toward greater equity in the knowledge of and approach to diseases that affect the whole population or only part of it [15,20,22].

Knowing that chronic conditions entail a loss of quality of life, it should be taken into account that this aspect is more accentuated in women. This is due, in part, to their greater longevity and, consequently, a greater risk of presenting chronic conditions but also derives from a worse perception of their health status [9,15]. One factor that may play a role is the caregiving responsibilities that women have traditionally carried out. Increased life expectancy has meant a greater need for care [13], which cannot always be covered by the care provided by the health and social services system [6,23]. Informal care is provided by the family environment and the most immediate social network, generally women, without them receiving a professional economic benefit in return [23,24]. Taking on the care of a dependent person constitutes both a risk factor for the caregiver’s health and quality of life. This is related to higher levels of stress and depression and worsened subjective well-being and physical health, and for a reduced likelihood of engaging in self-preventive actions and seeking healthcare [24].

In Catalonia (Spain), the incorporation of a sex and gender perspective into health has spread among the sector’s strategic plans over the last few years, as well as in clinical guidelines and diagnosis and treatment manuals [21]. By emphasising the importance of providing evidence without extrapolating case-based reasoning from one sex to the other, we are moving towards a person-centred model of care [25,26]. As is the case in other parts of Central Catalonia, the population of the counties of Bages and Moianès show a sustained increase in population ageing and over-ageing, a fact that corresponds with its characterisation as a rural region [27]. Despite an expected high prevalence of chronic conditions and situations of chronic multimorbidity, this context has not been analysed from a gender perspective.

Furthermore, although the prevalence of chronic conditions and multimorbidity is expected to be higher as age progresses, these types of diseases are not exclusive to older ages. The literature has focused less on understanding how this type of conditions affect younger populations [5]. It is important to consider both a gender and age perspective in order to understand the reality of this territory, which could guide current initiatives in this area, as well as in the adoption of new strategies for the management of chronic conditions and innovative procedures or treatments [28].

The aim of this study is to describe, from a sex and gender perspective, the situation and distribution of the prevalence of chronic conditions in the counties of Bages and Moianès between 2018 and 2021, as well as to analyse the associations of the diseases analysed according to sex and age.

## 2. Materials and Methods

A retrospective descriptive cross-sectional study was carried out, according to the STROBE criteria [29], using data from the Information Systems of the Technical Department of the Territorial Management of Central Catalonia of the Catalan Health Institute (ICS) [30]. The ICS is the largest provider of the Catalan Health Service, although it is not the only one. The data analysed corresponded to people of any age assigned to ICS centres in the counties of Bages and Moianès between 2018 and 2021, presenting at least one of the chronic conditions chosen for the study.

The region of Bages–Moianès, with an area of 1430 km^2^, is eminently rural, as it has only two urban centres (Manresa and Sant Joan de Vilatorrada) [27]. It is part of Central Catalonia, one of the seven regions that make up the Catalan healthcare system. This is configured on the basis of a public model of universal care and provided from citizens’ taxes [31]. Of the 193,880 people in the region in 2021 [32], the population assigned to ICS centres was 163,024 people (approximately 85% of the total).

### 2.1. Variables

Understanding a chronic condition as one that is usually of long duration, slowly progressive and non-communicable among people [2,3,26], we selected a total of 26 diseases as dependent variables. These were chosen from the most frequent and impactful chronic conditions encountered in primary care. We included diseases like ADHD and ASD, acknowledging their chronic nature despite them not fitting the conventional definition of slow progression (Appendix A, Table A1). These were grouped into 7 categories for the analyses: neurodegenerative diseases, neurodevelopmental disorders, mental disorders, cardiovascular diseases, respiratory diseases, metabolic diseases and chronic pain. Multimorbidity was defined as experiencing two or more chronic conditions.

Most of these diseases are usually treated in primary care in Catalonia with occasional support, depending on the evolution of the disease, using specialised hospital care [33]. In the categories of neurodegenerative diseases and neurodevelopmental disorders, specialised care is usually more present in the whole approach. The inclusion of these two categories also responds to the scope of a broader project in which this study is framed, the PECT BAGESS (Bages Territorial Specialisation and Competitiveness Project). This one focuses on chronic conditions and dependence in the region of Bages, with the participation of several entities of the region with areas of expertise focused on specific populations [28].

The independent variables were age and sex. For this study, the sex variable is used for the analyses since the database does not have information related to the gender of the population studied, such as gender roles or identities [34,35]. Therefore, a gender perspective is applied to the interpretations that can be derived from the results.

### 2.2. Data Analysis

Categorical variables were described as percentages and 95% confidence intervals (95% CI). We estimated the prevalence of chronic conditions with a corresponding 95% CI, stratifying by sex and age (≤15 years, 15–45 years, 45–75 years, ≥75 years), during the study period (2018–2021). At the same time, prevalence was estimated for the years 2018 and 2021 to determine the changes at these two points in the period studied.

A logistic regression model was performed to estimate the associations between the chronic conditions studied (categorised into the seven groups) and age groups, stratified by sex. The study population had to present at least one of the chronic conditions analysed. The results were presented as odds ratio (OR) with a 95% CI. At the same time, the models were adjusted taking into account women and men separately, in order to observe the nature of the associations in a differentiated manner. Confidence intervals were used to test for statistically significant differences.

All analyses were performed using version 4.1.2 of the statistical program R.

## 3. Results

### 3.1. Population Profile

Between 2018 and 2021, data were retrieved for a total of 75,936 people with at least some chronic condition, representing 46.6% of the population assigned to ICS health centres in the Bages–Moianès area (Table 1). The prevalence of all chronic conditions grouped together was significantly higher in women (50.1%, 95% CI 49.8–50.5) than in men (43%, 95% CI 42.7–43.4). The mean age was 56.7 years (*SD* = 21.4). The age group with the highest prevalence of chronic conditions was people over 75 years of age (93.1%, 95% CI 92.7–93.5). A total of 22.6% of the population analysed had multimorbidity (almost half of those with chronic conditions), with a mean age of 65.9 years (*SD* = 17.7). Multimorbidity was also higher among women (25.8%, 95% CI 25.5–26.1) compared to men (19.4%, 95% CI 19.1–19.7) and with a higher prevalence in the over-75 age group (73.8%, 95% CI 73.1–74.4).

The prevalence of chronic conditions remained significantly higher in women, with more multimorbidity than single chronic conditions, compared to the distribution in men. The mean age in men was slightly lower than in women in both cases. Overall, the prevalence of chronic conditions was higher in all age groups in females, except in children younger than 15 years and in single chronic conditions in those older than 75 years (Table 2).

### 3.2. Distribution of the Prevalence of Chronic Conditions

Overall, the most prevalent diseases among the studied population were anxiety (19.1%, 95% CI 18.90–19.28), hypertension (18.03%, 95% CI 17.84–18.21), pulmonary diseases except COPD (9.6%, 95% CI 9.41–9.69), osteoarthritis (8.5%, 95% CI 8.33–8.60), depression (8.2%, 95% CI 8.07–8.34) and type 2 diabetes mellitus (7.2%, 95% CI 7.08–7.33) (Appendix B, Table A2).

Among these, statistically significant differences were observed according to sex, especially in the prevalence of anxiety, osteoarthritis and depression, where the values for women were double those for men. Also observed was a significantly higher prevalence in women compared to men of Alzheimer’s disease, dementia and all diseases related to chronic pain. Conversely, cases of autism spectrum disorder (ASD), attention deficit/hyperactivity disorder (ADHD), chronic obstructive pulmonary disease (COPD) and ischemic heart disease presented higher values among men than among women.

Globally, the prevalence of the analysed conditions grew between 2018 and 2021 in both women and men (Appendix B, Table A3 and Table A4).

Grouped according to the type of chronic conditions (Table 3), mental disorders (24.1%, 95% CI 23.8–24.3) and cardiovascular diseases (19.9%, 95% CI 19.7–20.1) were the most prevalent groups. Among women, the highest prevalence remained in mental disorders (31.2%, 95% CI 30.8–31.5) and cardiovascular diseases (19.8%, 95% CI 19.5–20.1), followed by diseases related to chronic pain (13.7%, 95% CI 13.5–13.9). In contrast, among men, the prevalence of cardiovascular diseases (20.1%, 95% CI 19.8–20.4) was higher than that of mental disorders (16.9%, 95% CI 16.6–17.2), with respiratory diseases following this in terms of prevalence (11.1%, 95% CI 10.9–11.3) and with equivalent values among women.

Stratifying by age, the prevalence of chronic conditions was at its peak after the age of 75 in both sexes. However, the prevalence of mental disorders was quite high in the 15–45 years age group in women (25.5%, 95% CI 24.98–26.0) and in men (15.7%, 95% CI 15.3–16.1) compared to other types of chronic conditions. Mental disorders and chronic pain-related conditions were also significantly more prevalent among women than men at all ages. Cardiovascular diseases were more prevalent at younger ages in men than in women, despite presenting a similar prevalence at 75 years of age and older. Neurodevelopmental disorders were primarily concentrated among young people under 45 years of age and significantly more prevalent among men. Except for this type, all diseases showed a positive gradient according to age.

### 3.3. Associations between Chronic Conditions, Age and Sex

The associations of presenting each type of disease grouped by sex and by age (adjusted by sex) were estimated using a logistic regression model (Table 4). In the population with at least one chronic condition, being male was associated with a higher probability of having cardiovascular diseases (OR = 1.58, 95% CI 1.52–1.64), neurodevelopmental disorders (OR = 1.97, 95% CI 1.88–2.13) and metabolic diseases (OR = 1.63, 95% CI 1.56–1.69), while it was a protective factor in neurodegenerative diseases (OR = 0.83, 95% CI 0.75–0.91), mental disorders (OR = 0.36, 95% CI 0.35–0.37) and chronic pain (OR = 0.46, 95% CI 0.45–0.48).

The effect of age was observed to be similar among both sexes, except for respiratory diseases. In this case, women began to present these diseases more markedly at younger ages, showing a negative gradient. At the same time, a difference was observed in the 45–74 years age group (OR = 1.12, 95% CI 1.05–1.19) compared to the same group in the case of men, where it was less likely (OR = 0.70, 95% CI 0.66–0.75). Being older was a protective factor in the case of neurodevelopmental disorders, contrary to what was observed regarding neurodegenerative, cardiovascular, metabolic and chronic pain diseases. No gradient determined by age was observed in the case of mental disorders, beyond the fact that being younger than 15 years of age was a protective factor. However, the group aged 15–45 years had a higher odds of presenting mental disorders than the other groups.

## 4. Discussion

This study has made it possible to describe the prevalence of various chronic conditions in the region of Bages–Moianès between 2018 and 2021, exploring the associations that occur in relation to age and sex. Of the total number of people assigned to ICS centres in this region, almost half had at least one of the chronic conditions analysed in this study. The prevalence among women was higher of both single chronic conditions and in cases of multimorbidity, except in children under 15 years of age. Overall, the age group with the highest prevalence of chronic conditions was found to be over 75 years of age, with a positive association with increasing age. At the same time, the prevalence of chronic conditions grew in the period studied, in line with the increasing trend in recent years [5].

The results obtained regarding the prevalence of chronic conditions and multimorbidity are consistent with the existing literature in an international context. Special emphasis is placed on the increase in chronic conditions in older adults (65 years of age and older), but this trend is also evident in younger people [5,36,37]. At the local level, chronic condition values slightly higher than those estimated in the 2021 Catalan Health Survey have been observed, although the distribution according to age and sex and the increasing trend coincide [38,39,40].

As for the types of diseases, the results obtained are in line with recent studies, identifying anxiety, hypertension, respiratory diseases, osteoarthritis, depression and type 2 diabetes as the most prevalent chronic conditions [37,41]. The differences according to sex by disease group obtained correspond to the expected values [39,40]. It is noteworthy that the highest number of cases of neurodevelopmental disorders is concentrated in males in the younger age groups, pointing to potential underdiagnoses of this type of disorders in females because of a gender bias due to sets of symptoms that are not considered typical at early ages [15,42,43]. This is consistent with differences in the number of visits to child mental health services in the Central Catalonia region in the years 2019–2020, which were markedly higher in males [40].

Remarkably, mental health issues prevalence has been high in both young people and women in general [39]. In a post-COVID-19 pandemic context, concern over the mental health of the population and specifically of youth is on the rise, especially given the effects of the period of confinement in 2020 [39,44]. Despite the decrease in mental health visits in this period and the consequent reduction in diagnoses, visits for anxiety as of July 2020 exceeded those of the previous year in Catalonia [44].

Considering that the Catalan public health system still has limited resources to address mental health, the results of this study reinforce the urgency of adopting action plans that take into account the importance of this area of health [45]. At the same time, it has been shown that the risk of suffering mental disorders such as anxiety and depression increases in cases of chronic conditions [5]. For this reason, a comprehensive approach to mental healthcare that is not based solely on medicalisation [17] is key to avoiding an increase in the burden of disease and multimorbidity.

Although the bulk of chronic conditions are concentrated after the age of 45, the results obtained on certain groups of diseases at younger ages (especially respiratory diseases, mental disorders and neurodevelopmental disorders) should not be underestimated. Knowing that at older ages chronic conditions and comorbidities are more likely, it is crucial to work on prevention and health promotion at all stages of life [46]. Providing health education and considering the experiences of chronic conditions in young people can help prevent disease overload [47], ensuring a higher quality of life in later life.

Furthermore, analysing the associations between chronic conditions and sex and age allowed us to estimate that the risk of presenting neurodegenerative diseases, mental disorders and chronic pain was lower in men. Positive age gradients have been observed in most conditions, except neurodevelopmental and respiratory disorders. Diagnoses mostly occurring at young ages in patients with neurodevelopmental disorders may elucidate this fact [43,48]. A high predisposition to presenting respiratory diseases in childhood (e.g., asthma or bronchitis) would explain the results obtained [49]. This could also suggest an immune delay in children due to the use of masks caused by the COVID-19 pandemic. This widespread adoption of mask-wearing by children as a public health measure may have influenced the development of their immune systems and respiratory health by limiting their exposure to common pathogens (called immune debt) [50]. In the case of mental disorders, it has been observed that exposure was higher in the 15–45 years age group in both sexes compared to other age groups, which reinforces the need to address these types of chronic conditions early.

Overall, the results point to a greater number of chronic conditions in women, especially with multimorbidity. However, the chronic care system in Catalonia has been characterised as reactive, paternalistic, fragmented and focused on certain conditions, especially cardiovascular, respiratory and metabolic ones [38]. Although the Catalan health system is currently adapting to a more person-centred approach [33,45], it still does not have sufficient resources to guarantee a comprehensive and continuous approach to conditions that involve a lower fatality but that affect to an equal or greater extent the quality of life of the people who suffer from them [15]. Chronic pain and mental health conditions, which affect women to a greater degree [14,51], are particularly responsive to this casuistry (i.e., a lack of a continuous approach or standardised and planned attention), requiring affected individuals to seek health services on demand. Although the care that these cases may require does not match up to a level of care that would classify them as complex chronic patients, it is important to have a standardised approach to them. A recent qualitative study drew on the perception of people affected by chronic conditions in Catalonia, pointing out the factor of feeling that they were a burden by wasting the time of professionals [33], a feeling that could be worsened by the lack of follow-up procedures and a continuous approach.

Despite advances in adapting and efforts to adapt the healthcare system, it still has a primarily biomedical and androcentric approach. As a result, care in cases presenting pain, emotional or physical discomfort among women tends to be slower than among men [17,33], given the historical neglect of women’s symptoms and experiences, which has influenced the absence of specific diagnoses and differential protocols [14,15]. This delay in undergoing diagnostic tests and treatments also leads to a worse self-perceived health status in women [52,53]. On the other hand, the gender role associated with caregiving (especially informal) must be taken into account, contributing to an overload of care for women. This would be one of the factors that could also explain the age differences in the presence of certain chronic conditions, due to postponing seeking care for their own health problems, prioritising that of their dependents [54].

Although the prevalence of chronic conditions increases with age, and women are more affected, the results showed that chronic conditions do not only affect older adults. It is noteworthy that among the 15–45 years group, the prevalence of a single chronic condition was 22.9% for men and 25.9% for women. As mentioned before, one of the most alarming age-related findings was the prevalence of mental health issues among young people. In relation to these findings, a recent study showed that diagnoses of depression and mood disorders in primary care in Catalonia increased by 86.6% between 2017 and 2022. In particular, there was a significant increase in diagnoses among young people (0–34 years) in 2022 [55]. The need to devise strategies to tackle mental health issues in the Catalan context was emphasised. Understanding how individuals’ behaviours align with societal norms, their inclination to seek assistance and their coping mechanisms, especially when stratified by age, sex and gender, can contribute to developing more precise strategies.

In terms of resources, it should be borne in mind that the Catalan health system, especially primary care, currently has a very high healthcare burden. This could be a factor in the difficulty of making changes towards a person-centred model in all aspects and areas of health [56]. For this reason, it is crucial that not only sufficient resources are available but that all teams of professionals are trained and sensitised in gender issues to deal with increasing chronic conditions, taking into consideration all stages of life, as well as change management and resilience tools. Similarly, university and professional training plans in this sector should include these aspects [57].

As stated in the Catalan Health Plan 2021–2025, the incorporation of the gender perspective at all levels of health services is necessary [45]. This stands as the initial step towards establishing specific strategies adapted to the contexts of the services. At the same time, it is necessary to take into account all the stakeholders that will guarantee this change, from the professional sphere to the affected persons, caregivers and environment [28,33]. This can be especially relevant for future public health programmes and community health initiatives. Future studies should take into account other social determinants of health, in order to be able to analyse, prevent and act on those cases where various axes of inequity intersect [52,53]. Given the complexity of approaching chronic conditions, this topic cannot be dealt with without taking into account all the biopsychosocial, cultural and environmental conditioning factors that come into play [13,14,15].

### Strengths and Limitations

In a context of growing interest in the impact of chronic conditions, this study is among the first to describe chronic conditions in the Bages–Moianès area from a gender perspective. Adopting this approach has made it possible to accurately distinguish some of the main chronic conditions, characterising their distribution by sex and age.

As for the main limitations, the study design does not allow causality to be determined. Nevertheless, this study sought to preliminarily identify the distribution of chronic conditions in the region, with a view to guiding future actions and studies and serving as a foundation for the PECT BAGESS, an initiative aimed at enhancing the area’s specialisation and competitiveness in health-related fields [28]. Secondly, the diversity of the analysed population might be less diverse than in greater urban areas since it tackles a predominantly rural and ageing population. Even so, the results obtained are consistent with other local studies. Further analysis adopting other methodologies would be useful to contrast these findings considering diverse axes of inequality. Additionally, there is a potential information bias for certain chronic conditions, which could be due to the lack of introduction of diagnoses and also due to the dual public–private coverage that a percentage of the population has. However, the large sample size of this study could minimise possible errors in the estimates proposed. In relation to the above, this study included a list of diseases that may underplay certain chronic conditions in terms of the definition of diagnostic codes. In any case, the diversity of diseases covered provides a cross-sectional view of the situation of chronic conditions in the region.

## 5. Conclusions

The prevalence of chronic conditions shows an upward trend in the area of Bages–Moianès, as is being observed globally. Noting an increase in prevalence between 2018 and 2021, almost half of the population assigned to ICS centres in this region presented at least one of the chronic conditions analysed in this period. The study has identified a higher prevalence of chronic conditions (both single conditions and multimorbidity) among women and older adults. Considering the complexity of approaching chronic conditions, a gendered approach must be prioritised in order to guarantee accurate and careful attention to diversity. Future analyses with a gender perspective and that take into account the social determinants of health are needed to determine how the axes of inequity affect the processes of chronic conditions and the care they require.

## Figures and Tables

**Table 1 ijerph-21-00152-t001:** Prevalence with 95% CI of chronic conditions and multimorbidity by sex and age.

	Population (*n* = 163,024)	Prevalence of Chronic Conditions (*n* = 75,936)	Prevalence of Multimorbidity (*n* = 36,888)
Total	100%	46.6% (46.3–46.8)	22.6% (22.4–22.8)
Sex			
Women	81,861 (50.2%)	50.1% (49.8–50.5)	25.8% (25.5–26.1)
Men	81,163 (49.8%)	43% (42.7–43.4)	19.4% (19.1–19.7)
Age–mean (SD)	N/A	56.7 (21.4)	65.9 (17.7)
[0–15)	23,743 (14.6%)	15.0% (14.5–15.5)	1.6% (1.5–1.8)
[15–45)	57,618 (35.3%)	31% (30.6–31.3)	7% (6.8–7.2)
[45–75)	64,807 (39.8%)	59.9% (59.5–60.3)	30.9% (30.6–31.3)
≥75	16,565 (10.2%)	93.1% (92.7–93.5)	73.8% (73.1–74.4)

Age is presented as mean and standard deviation for the population with chronic conditions and multimorbidity and as relative frequency for every range. Prevalence of chronic conditions includes multimorbidity. Multimorbidity refers to two or more chronic conditions.

**Table 2 ijerph-21-00152-t002:** Prevalence with 95% CI of chronic conditions among women and men by age.

	Single Chronic Condition		Multimorbidity	
(*n* = 163,024)	Women (*n* = 19,895)	Men (*n* = 19,153)	Women (*n* = 21,131)	Men (*n* = 15,757)
Total	24.3% (24.01–24.6)	23.6% (23.3–23.9)	25.8% (25.5–26.1)	19.41% (19.1–19.7)
Age–mean (*SD*)	47.4 (20.3)	45.6 (20.9)	66.6 (17.5)	63.9 (17.8)
[0–15)	11.0% (10.5–11.6)	15.6% (14.9–16.2)	1.1% (0.9–1.3)	2.1% (1.83–2.35)
[15–45)	25.9% (25.4–26.4)	22.1% (21.7–22.6)	7.9% (7.7–8.3)	6.0% (5.7–6.3)
[45–75)	29.6% (29.2–30.2)	28.3% (27.8–28.8)	34.3% (33.8–34.8)	27.7% (27.2–28.2)
≥75	18.0% (17.3–18.8)	21.5% (20.5–22.5)	76.1% (75.3–76.9)	70.1% (69.0–71.2)

Age is presented as mean and standard deviation for the population with chronic conditions and multimorbidity and as relative frequency for every range. Multimorbidity refers to two or more chronic conditions.

**Table 3 ijerph-21-00152-t003:** Prevalence with 95% CI of chronic conditions grouped by sex and age.

	ND	NDDs	MHD	Respiratory	CVD	Metabolic	Chronic Pain
Total (*n* = 163,024)	1.34% (1.3–1.4)	2.4% (2.3–2.5)	24.1% (23.8–24.3)	11.2% (11.0–11.4)	19.9% (19.7–20.1)	7.5% (7.3–7.6)	10.1% (9.9–10.2)
Women (*n* = 81,861)	1.8% (1.7–1.9)	1.5% (1.4–1.6)	31.2% (30.8–31.5)	11.3% (11.1–11.5)	19.8% (19.5–20.1)	6.6% (6.4–6.8)	13.7% (13.5–13.9)
Age							
[0–15)	-	3.1% (2.8–3.4)	1.8% (1.5–2.0)	7.7% (7.2–8.2)	0.18% (0.1–0.3)	0.24% (0.2–0.4)	0.01% (0.01–0.08)
[15–45)	-	2.4% (2.9–2.5)	25.5% (24.98–26.0)	8.4% (8.1–8.8)	1.3% (1.1–1.4)	0.8% (0.7–0.9)	1.7% (1.6–1.9)
[45–75)	0.60% (0.5–0.7)	0.55% (0.5–0.6)	42.0% (41.5–42.6)	12.6% (12.2–12.96)	24.8% (24.3–25.3)	8.7% (7.96–8.6)	19.4% (18.9–19.8)
≥75	12.2% (11.6–12.9)	0.16% (0.09–0.26)	45.7% (44.7–46.7)	19.2% (18.4–19.9)	77.1% (76.3–77.9)	24.6% (23.8–25.4)	44.3% (43.4–45.3)
Men (*n* = 34,910)	0.91% (0.8–0.97)	3.4% (3.2–3.5)	16.9% (16.6–17.2)	11.1% (10.9–11.3)	20.1% (19.8–20.4)	8.4% (8.2–8.6)	6.39% (6.22–6.56)
Age							
[0–15)	-	7.6% (7.1–8.1)	1.1% (0.9–1.3)	9.99% (9.5–10.5)	0.13% (0.08–0.22)	0.15% (0.10–0.25)	-
[15–45)	-	5.4% (5.1–5.6)	15.7% (15.3–16.1)	8.1% (7.8–8.4)	2.1% (1.9–2.2)	0.96% (0.85–1.08)	0.91% (0.81–1.03)
[45–75)	0.61% (0.53–0.7)	0.63% (0.55–0.72)	22.3% (22.5–23.4)	11.2% (10.8–11.5)	31.98% (31.5–32.5)	13.4% (13.0–13.8)	9.6% (9.3–9.97)
≥75	8.0% (7.5–8.8)	0.2% (0.13–0.4)	21.6% (20.6–22.6)	26.3% (25.2–27.4)	78.0% (76.98–79.0)	31.5% (30.4–32.7)	26.5% (25.4–27.6)

CVD = cardiovascular disease; MHD = mental health disorder; ND = neurodegenerative disease; NDDs = neurodevelopmental disorders.

**Table 4 ijerph-21-00152-t004:** Logistic regression analysis of associations between chronic conditions and age groups, stratified by sex.

(*n* = 75,936)	ND	NDDs ^†^	MHD	Respiratory	CVD	Metabolic	Chronic Pain
	OR (CI 95%)						
Sex (men)	0.83 * (0.75–0.91)	1.97 * (1.82–2.13)	0.36 * (0.35–0.37)	0.96 (0.93–1.00)	1.58 * (1.52–1.64)	1.63 * (1.56–1.69)	0.46 * (0.45–0.48)
Women							
Age							
[0–15)	-	[R]	0.17 * (0.15–0.21)	5.82 * (5.16–6.58)	-	0.05 * (0.03–0.07)	-
[15–45)	-	0.51 * (0.44–0.60)	3.21 * (3.02–3.41)	1.71 * (1.59–1.84)	0.01 * (0.01–0.01)	0.07 * (0.07–0.09)	0.06 * (0.05–0.07)
[45–75)	0.06 * (0.05–0.07)	0.04 * (0.04–0.06)	2.03 * (1.94–2.14)	1.12 * (1.05–1.19)	0.15 * (0.14–0.16)	0.46 * (0.43–0.49)	0.51 * (0.48–0.53)
≥75	[R]	0.006 * (0.004–0.011)	[R]	[R]	[R]	[R]	[R]
Men							
Age							
[0–15)	-	[R]	0.22 * (0.18–0.26)	2.97 * (2.68–3.29)	-	0.01 * (0.01–0.02)	-
[15–45)	-	0.59 * (0.53–0.66)	4.07 * (3.78–4.38)	1.34 * (1.24–1.45)	0.01 * (0.01–0.01)	0.08 * (0.07–0.09)	0.09 * (0.07–0.10)
[45–75)	0.10 * (0.08–0.12)	0.02 * (0.02–0.03)	2.24 * (2.10–2.40)	0.70 * (0.66–0.75)	0.26 * (0.24–0.28)	0.67 * (0.63–0.72)	0.54 * (0.50–0.57)
≥75	[R]	0.004 * (0.002–0.006)	[R]	[R]	[R]	[R]	[R]

* = *p* < 0.001; CVD = cardiovascular disease; MHD = mental health disorder; ND = neurodegenerative disease; NDDs = neurodevelopmental disorders; [R] = reference. ^†^ = Given a highly concentrated distribution of cases at young ages, the reference category was modified to run the model and correct for associations in NDDs.

## Data Availability

Access to the data used in this article is limited to the Catalan Health Institute and to the researchers participating in this study due to the sensitive nature of the personal data and the requirements set forth by Spanish and European regulations. Data sharing is not applicable to this article.

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
