# Peer review of "A Sex- and Gender-Based Approach to Chronic Conditions in Central Catalonia (Spain): A Descriptive Cross-Sectional Study"

_ijerph, 2024, doi:10.3390/ijerph21020152_

Round 1

Reviewer 1 Report

Comments and Suggestions for Authors

This study makes an important contribution in it’s efforts to identify the prevalence of chronic health conditions for people living in the Bages-Moianes region of Catalonia Spain and specifically consider differences in single diagnoses multimorbidity based on patients’ sex and age. The findings present a solid case for the importance of healthcare systems to adopt perspectives and approaches that take women’s unique health needs into greater account. I greatly enjoyed reading your work. I have points  and recommendations mainly in regard to clarifying the manuscript’s meaning and/or phrasing.

Introduction

1. The point that more research is needed to understand the differential impact on men and women from chronic health conditions is highly warranted. That said, there is research out there and likely a bit more than is implied by the start of the second paragraph (See Adab-Diez et al., 2014; Buttorff et al., 2017). It could add precision to the manuscript to acknowledge more of the work that has been done on this topic before this study. That said, the argument developed later in this paragraph and the next is highly relevant.

2. Also related to the second paragraph, and specifically the phrasing “the existence of studies on the differential impact on men and women…”, this reads as if implying ‘the impact these diseases have on men and women’ (i.e., outcomes from the diseases) rather than the prevalence of them. Can you clarify the wording here?

3. An implication from the introduction (and how it’s structured) is that the focus of this study is on gender and sex differences in regard to the prevalence of chronic conditions, including multimorbidity. There’s a brief mention of the relationship between age and chronic disease at the start of the manuscript, though by the end of this section, this part of the rationale has veered off a bit. It could strengthen the rationale to refer back to the argument for examining differences via age when describing the study aim. Just to reinforce that part of your research objective since more attention in this section has been on reasoning for examining sex differences.

Methods

3. Can you speak more on the criteria or decision-making process for the chronic diseases that were chosen? How many total diseases were considered that the 26 were selected from? Why those 26 specifically? Also, in relation to the description of a chronic condition as being “slowly progressive”, do all the 26 diseases in fit that criterion (especially the ones such as ADHD and ASD)? Can you speak more on the inclusion of these, especially if they are not considered always progressive in nature.

4. Later in the section on “Variables” when describing the project scope, the wording reads a little vaguely. It would help to be a little more specific here. What is the scope of the project? How does it relate to or provide an explanation for examining neurodegenerative diseases and neurodevelopmental disorders along with physical conditions? That was something I was especially curious about, regarding the decision to examine these different categories of chronic conditions (neuro and physicl). Also, please clarify what you mean by ‘whole approach’ and also the mention of “dependence” in this section. Do you mean patients’ dependence on the healthcare system? I didn’t get a sense of that notion being examined in this study, rather the aim was to report on prevalence of diagnoses. The last sentence of the second paragraph under Variables was a little confusing. Overall, this section led me to question if you meant this study had been developed from a larger project or broader study. If so, please elaborate.

Data Analysis

5. It could add clarity in the data analysis section to include that your use of CI was also the method used to test for statistically significant differences.

Results

6. When first mentioning “Prevalences of chronic conditions…” on p 4, having a qualifier like all together or as a whole (etc.) after “chronic conditions” could help clarify that in this part, you are referring to all conditions grouped together.

Discussion

7. Can you clarify what you mean about children’s use of masks during the COVID-19 pandemic was consolidated?

8. The start of paragraph 9 in this section is a little vague. Since it’s starting a new idea/paragraph, it would help readers to specify again (or remind them) what you mean by ‘this casuistry’. What specifically about the Catalan health system?

9. The first sentence of the last paragraph in this section is vague and/or the specific idea to be conveyed in this sentence is unclear. I’d consider rewording it and/or making two sentences.

Table

10. Table A1: For clarity and precision, I’d recommend using the full name when identifying Alzheimer's disease. Also, considering this disease is a form of dementia, consider another way to refer to the broader category you have for dementia. (To avoid implying they are separate conditions.) Meaning, reflect on how your labeling can imply your examining other forms of dementia.

Thank you for the opportunity to read your work!

Comments on the Quality of English Language

See comments above regarding quality of English language. 

Reviewer 2 Report

Comments and Suggestions for Authors

The paper aimed to describe, from a sex and gender perspective, the situation and distribution of the prevalence of chronic conditions in the counties of Bages and Moianès between 2018 and 2021, as well as to analyse the associations of the diseases analysed according to sex and age.

There are some aspects that definitely need to be revised, as follows:

1. “Social and health progress has allowed life expectancy to increase over the past few decades worldwide (1,2)“. It seems necessary to arrange the references in all manuscript text according to the required citation style.

2. Dependent and independent variables need to be defined very precisely. This design of the study is incomprehensible.

3. Table 1 shows prevalence as a percentage. Prevalence must be reported for 10 000 or 100 000 inhabitants.

4. As the authors write, the logistic regression model was not sex-specific. The analysis was simply carried out in different sub-groups.

5. What were the eligibility criteria for logistic models?

Reviewer 3 Report

Comments and Suggestions for Authors

Interesting data from primary care clinics in the Bages-Moianes region of Spain are reported with a focus on age and gender variations in the chronic conditions reported  The sample includes both males and females age <15 to >75.  The chronic conditions are divided into 7 categories, some of which primarily involve those of a younger age while others involve primarily older adults. At this level of generality, there are probably few surprises in the results. The authors' interpretation of the results raise some questions, however.  For example, is anxiety really more prevalent in women or is it more readily reported by women or more readily noted in women by primary care providers? The same questions may apply to pain which is by necessity based upon patient report as opposed to diagnostic procedures. The authors suggest this data may be used for healthcare planning but offer few specific suggestions or additional uses.  This is a lost opportunity to demonstrate the value of the data analyzed.  In addition, several comments in the manuscript  raise issues that are not addressed in the discussion.  These involve the extent to which the chronic conditions reported are really "burdensome" as opposed to manageable, which is often the case. It is also important to discuss the roles of age and gender in the occurrence of these conditions which is not done. This type of discussion in relation to the results would increase both the meaning and the value of the analysis. 

There are also some technical issues that need to be addressed. First, what do the authors mean by a "gender sensitive approach"?  How would this be done in this and other settings.  What would be the anticipated effect on these statistics? Second, the use of the term "older adult" is preferable to use of the term "elderly" and this should be adjusted. Third, the exclusion of women from many research studies is no longer the problem it once was and this comment should be edited accordingly.  Finally, the meaning of the sentence found in lines 334-337 needs to be made clearer.

Round 2

Reviewer 2 Report

Comments and Suggestions for Authors

The authors did not change the manuscript according to my original notes.
Therefore, I recommend to change the paper based on my original suggestions.

Warm regards
